# Acid Sphingomyelinase Deficiency: Sharing Experience of Disease Monitoring and Severity in France

**DOI:** 10.3390/jcm11040920

**Published:** 2022-02-10

**Authors:** Wladimir Mauhin, Raphaël Borie, Florence Dalbies, Claire Douillard, Nathalie Guffon, Christian Lavigne, Olivier Lidove, Anaïs Brassier

**Affiliations:** 1Service de Médecine Interne, Centre de Référence Maladies Lysosomales, Groupe Hospitalier Diaconesses Croix Saint-Simon, 75020 Paris, France; WMauhin@hopital-dcss.org; 2Service de Pneumologie A, Hôpital Bichat, 75018 Paris, France; raphael.borie@aphp.fr; 3Unité de Recherche, INSERM, Unité 1152, Université Paris Diderot, 75018 Paris, France; 4Institut de Cancéro-Hématologie, CHU Morvan, 29200 Brest, France; florence.dalbies@chu-brest.fr; 5Centre de Référence des Maladies Héréditaires du Métabolisme, Avenue Avinée, Hôpital Jeanne de Flandres, CHU Lille, 59000 Lille, France; claire.douillard@chu-lille.fr; 6Centre de Référence Lyonnais des Maladies Héréditaires du Métabolisme, Hospices Civils de Lyon, HCL, 69677 Bron, France; nathalie.guffon-fouilhoux@chu-lyon.fr; 7Service de Médecine Interne et Immunologie Clinique, Centre de Compétence des Maladies Métaboliques Héréditaires, CHU Angers, 49933 Angers, France; ChLavigne@chu-angers.fr; 8Service de Pédiatrie et Maladies du Métabolisme, APHP Necker, 75015 Paris, France; anais.brassier@aphp.fr

**Keywords:** acid sphingomyelinase deficiency, ceramide, enzyme replacement therapy, morbidity, mortality, Niemann–Pick disease, olipudase alfa, recombinant human acid sphingomyelinase, sphingomyelin

## Abstract

Acid sphingomyelinase deficiency (ASMD) is a rare inherited lipid storage disorder caused by a deficiency in lysosomal enzyme acid sphingomyelinase which results in the accumulation of sphingomyelin, predominantly within cells of the reticuloendothelial system located in numerous organs, such as the liver, spleen, lungs, and central nervous system. Although all patients with ASMD share the same basic metabolic defect, a wide spectrum of clinical presentations and outcomes are observed, contributing to treatment challenges. While infantile neurovisceral ASMD (also known as Niemann–Pick disease type A) is rapidly progressive and fatal in early childhood, and the more slowly progressive chronic neurovisceral (type A/B) and chronic visceral (type B) forms have varying clinical phenotypes and life expectancy. The prognosis of visceral ASMD is mainly determined by the association of hepatosplenomegaly with secondary thrombocytopenia and lung disease. Early diagnosis and appropriate management are essential to reduce the risk of complications and mortality. The accessibility of the new enzyme replacement therapy olipudase alfa, a recombinant human ASM, has been expedited for clinical use based on positive clinical data in children and adult patients, such as improved respiratory status and reduced spleen volume. The aim of this article is to share the authors experience on monitoring ASMD patients and stratifying the severity of the disease to aid in care decisions.

## 1. Introduction

Acid sphingomyelinase deficiency (ASMD) is an inborn error of metabolism that leads to the accumulation of sphingomyelin in cells and tissues causing the clinical condition also known as Niemann–Pick disease type A, A/B and B (NPD) [1]. In ASMD, the enzymatic deficiency of the lysosomal acid sphingomyelinase (ASM), is caused by pathogenic variants of the sphingomyelin phosphodiesterase 1 gene (SMPD1; EC 3.1.4.12). Sphingomyelin is a major structural component of all plasma membranes. Cellular physiological function requires ASM to catalyze the hydrolysis of sphingomyelin to ceramide and phosphocholine (Figure 1). ASMD results in the progressive accumulation of sphingomyelin within all cells, but mainly in reticuloendothelial cells and so in spleen, liver, lung, bone marrow and lymph nodes, but also in neurons in neurovisceral forms. Sphingomyelinase enzyme activity generates ceramides that are bioactive sphingolipids that play a major role in inflammation. Ceramides act as second messengers transducing pro-inflammatory signals such as tumour necrosis factor-alpha [2,3]. Ceramides activate NFkB that in turn encodes for pro-inflammatory cytokines, such as interleukin (IL)-1 beta and IL-6 [4].

Numerous variants of the *SMPD1* gene, along with variability in residual ASM activity and other genetic/epigenetic factors, result in a spectrum of ASMD disease severity from a uniformly fatal form with death occurring by 3–4 years of age (ASMD type A or infantile neurovisceral ASMD previously known as NPD A) to chronic forms characterised by visceral (ASMD type B or chronic visceral ASMD, previously NPD B) and neurovisceral disease (ASMD type A/B chronic neurovisceral ASMD, previously NPD A/B) (Table 1) [5,6,7]. Of note, while Niemann–Pick disease type C and ASMD share several phenotypic features, they represent two distinct disease entities [8]. Reliable estimates of the birth prevalence of ASMD are currently lacking [9]; however, an estimated birth prevalence of 0.4–0.6 per 100,000 births has been reported [10]. In France, the birth prevalence of the chronic visceral, non-neurological form is 1/230,000 births and is recognised as being an underdiagnosed disease (clinical experience suggests 150 patients diagnosed in France over the last 30 years) [11].

The clinical phenotype and life expectancy of patients with ASMD type B appears to vary, with many adults reaching a normal lifespan, while others die prematurely from ASMD-related complications, such as respiratory insufficiency and liver disease [1,5,6,7,12]. Most patients with ASMD type B have interstitial lung disease with progressive impairment of pulmonary function, hepatosplenomegaly, haematological abnormalities (such as anaemia and thrombocytopenia) with bleeding, and an atherogenic lipid profile [7,9]. Diffuse interstitial lung disease with ground glass opacity, interlobular septal thickening, and intralobular lines, is common in ASMD type B [13]; however, symptoms vary from normal function through to respiratory failure [1,14]. Other common manifestations include liver disease (from persistent elevated transaminases [5] to cirrhosis [15] with possible liver failure [9]), heart disease (cardiac hypertrophy, valve regurgitation, conduction abnormalities, early myocardial infarction) [14], skeletal abnormalities, failure to thrive, and growth deficits in children which can persist into adulthood [7,16,17].

Data remain limited regarding predictors of disease-related morbidity, healthcare use, and lifestyle impact in patients with chronic ASMD; however, the physical, emotional, financial, and psychosocial burden of illness in patients with ASMD type B and ASMD type A/B is substantial [1,9,18,19]. McGovern et al. (2021) reported predictive factors of mortality that include both total splenectomy and spleen volume ≥15 multiples of normal (MN) at baseline [1]. Indeed, in an 11-year prospective natural history study of children and adults with ASMD, patients with a history of either severe splenomegaly or prior splenectomy had 10 times increased the likelihood of dying during follow up compared with those patients with moderate splenomegaly or intact spleens. On the contrary, liver volume of ≥2.5 MN was not shown to be a predictor of mortality. Interstitial lung disease was reported to worsen gradually with a mean diffusion capacity of carbon monoxide (DL_CO_) decreasing below 50% of the predicted value in adult patients. Atherogenic lipid profiles typically worsen with age in patients with ASMD type B [5], and lipid abnormalities may be associated with early coronary artery disease [20].

Due to the rarity and heterogeneity of the disease, there is a lack of robust quantitative data regarding the impact of the disease on patients’ and caregivers’ quality of life (QoL), compounded by the lack of adequate disease-specific instruments to measure QoL [9].

## 2. Clinical Study Evidence for the Use of Olipudase Alfa in the Non-Neurologic Manifestations of ASMD

Knockout mouse models have demonstrated that enzyme replacement therapy (ERT) is likely to be a useful therapeutic approach for patients with non-neurological ASMD, with a strong response to ERT in the major organs involved in the disease (liver, spleen, and lung) [21]. For the neurological manifestation of ASMD, ERT is unlikely to evoke any therapeutic response in the central nervous system since it does not cross the blood–brain barrier. Importantly, progressive dose escalation appeared essential to prevent the possibility of a deadly cytokine storm due to the abrupt release of high levels of ceramide [22].

Olipudase alfa is currently being investigated as an ERT for the treatment of the non-neurological manifestations of ASMD [21,23]. Of note, olipudase alfa is the first and only investigational ERT in late-stage development for ASMD. Olipudase alfa acts by targeting the underlying metabolic defect by supplementing the deficient enzyme activity [24].

A Phase 1A study assessed the infusion of a single dose (0.03–1.0 mg/kg) of olipudase alfa in 11 patients with ASMD type B [25]; a maximum starting dose of 0.6 mg/kg was identified which supported a dose-escalation strategy for the gradual clearance of accumulated sphingomyelin which was applied in a Phase 1B study. Of note, the aim of the dose-escalation process of olipudase alfa is gradual sphingomyelin debulking to prevent high ceramide release (Figure 1). Blood ceramide levels increased up to five-fold (both dose- and time-dependent), beginning 2–6 h after infusion, peaking between 24 and 48 h and returning to pre-infusion levels by Day 14. This led to systemic inflammation illustrated by high levels of C-reactive protein and bilirubin reported in patients who received the highest dosages of olipudase alfa (i.e., 0.6 and 1.0 mg/kg). No serious adverse drug reactions occurred during the study.

A Phase 1B study assessed safety and tolerability of olipudase alfa in five patients with ASMD type B reported no death, serious or severe adverse events (AEs), or AEs leading to discontinuation over the 26-week study period [26]; most AEs were mild (97%) and resolved without sequelae and all patients were successfully escalated to the maximum study dose of 3.0 mg/kg. A positive therapeutic response was also observed in liver sphingomyelin content, spleen and liver volumes, along with improvements in infiltrative lung disease, lipid profiles, platelet counts, and QoL. A sustained safety profile and continued improvements in clinically relevant parameters following up to 42 months of treatment in these patients has been reported [26,27,28].

A long-term safety study (NCT02004704) has demonstrated that olipudase alfa provides improvements in lipid profiles up to 42 months [28,29,30]; these include progressive reductions from baseline in pro-atherogenic lipid profiles (total cholesterol, low-density lipoprotein cholesterol [LDL-C], very low-density lipoprotein cholesterol, triglycerides) and progressive increases in anti-atherogenic markers (high-density lipoprotein cholesterol and apolipoprotein A-I).

Two separate clinical studies evaluating olipudase alfa for the treatment of chronic ASMD in adult and paediatric patients have also demonstrated positive results [27,31]. The Phase I/II open label, ascending dose, multicentre ASCEND-Peds study (NCT02292654) evaluated the safety, tolerability, and pharmacokinetics of olipudase alfa in 20 paediatric patients with chronic visceral ASMD over a 64-week study period [27]. A clinically meaningful improvement in pulmonary function assessed by the DLco and reduction in spleen size were reported—after 52 weeks of treatment, mean spleen volume decrease was shown to be 49%, while the nine patients who were able to perform the DLco test showed a mean increase of 33%. Most AEs were mild to moderate in severity, and no permanent treatment discontinuations were reported. The ongoing Phase II/III randomised, double-blind, placebo-controlled ASCEND (adults) study (NCT02004691) evaluated the efficacy, safety, and pharmacokinetics/pharmacodynamics of olipudase alfa in 36 adults with chronic visceral ASMD over a 52-week primary analysis period, which is now being followed by an extension period (all treated patients) of up to 4 years [31]. Patients received either olipudase alfa intravenous infusion or placebo every two weeks at an escalating dose from 0.1 mg/kg up to 3 mg/kg administered every two weeks. Figure 2 shows the dose-escalation strategy used in the ASCEND (adults) and ASCEND-Peds studies. The ASCEND (adults) study assessed two independent primary efficacy endpoints: DLco and spleen volume to reflect the separate critical manifestations of chronic visceral ASMD, with the trial outcome being deemed positive if one of these endpoints was met. Treatment outcomes favoured olipudase alfa (vs placebo) at Week 52 for spleen volume, percentage predicted DLco, liver volume and platelet levels. Overall, spleen volume was significantly decreased by 39.5% in the olipudase alfa arm compared with a 0.5% increase in the placebo arm (40% difference; *p* < 0.0001). In addition, percent change from baseline in percentage predicted DLco was +22% for the olipudase alfa arm at Week 52 compared with +3% for the placebo arm (19% difference; *p* = 0.0004). In addition, all treatment-related AEs were mild-to-moderate in severity and no patients discontinued due to AEs [19].

In 2015, the US Food and Drug Administration granted a breakthrough therapy designation to olipudase alfa based on the Phase 1b study data, an approach used to expedite the development and review of drugs intended to treat serious or life-threatening diseases and conditions [32]. Similarly, in 2016, olipudase alfa has gained orphan designation and ‘PRIority Medicines’ (PRIME) designation by the European Medicines Agency, along with being awarded the SAKIGAKE designation in Japan, in order to expedite the regulatory process [33].

## 3. Materials and Methods

Four expert meetings took place in Paris from 2019 to 2021, followed by several teleconferences, to gather expert opinion of French physicians with regards to the monitoring and severity stratification of children and adult patients with ASMD types B and A/B. All authors participated in the meetings apart from W.M. W.M., O.L., and A.B. have coordinated the process of early access to treatment in French ASMD patients. All co-authors have worked on this process.

## 4. French Expert Opinion: Monitoring of ASMD and Patient Severity Stratification

### 4.1. General Care of Patients with ASMD Type B Forms

Treatment goals for patients with chronic visceral ASMD should focus on reducing spleen and liver volumes, limiting hepatic fibrosis, decreasing cytopenia, improving liver function tests, and respiratory status in order to decrease morbidity and mortality [12]. However, the current management of ASMD type B has been limited to supportive care and palliation [9,13,34].

Regardless of aetiology, any respiratory failure should be treated with oxygen and respiratory rehabilitation [9,35,36]. Patients should also be encouraged to stop or avoid smoking (including second-hand exposure), to reduce the risk of lung and coronary disease, as well as cancer [13,37]. There is no specific recommendation but standard lipid-lowering agents, such as statins or ezetimibe, have shown benefits and are classically proposed to manage blood lipid abnormalities [9,13,38].

While there is no available treatment shown to improve bone density in patients with ASMD type B, the use of regular calcium and vitamin D supplementation to support healthy bone function is recommended [9]; of note, the use of bisphosphonates should be avoided given their potent inhibition of ASM [39].

Due to the low platelet count and splenomegaly, contact sports (particularly those with a risk of abdominal impact) should be avoided [9]. Splenectomy should be avoided as it has been demonstrated to increase the risk of lung disease and overall mortality [5].

Optimised immunisation in ASMD patient population is challenging because of a data gap in vaccine trials. Vaccinations against pneumococci, Haemophilus influenzae type B, meningococci, and influenza virus are recommended for patients with anatomic or functional asplenia [9,13,40]. While the use of the meningitis vaccine is mandatory in paediatric patients, its use in adult patients is also mandatory in patients with anatomic or functional asplenia [41]. In adult patients with platelets >30 G/L, vaccination against COVID-19 should now empirically be offered, as with all patients with a higher risk of severe COVID-19 due to chronic lung and/or liver disease [42].

### 4.2. Monitoring of Patients with ASMD Type B Forms

Scheduled patient monitoring remains essential to manage symptoms as early as possible, in order to minimise the risk of poor clinical outcomes. Based on our experience, a summary of key monitoring parameters along with the frequency of assessment for patients with ASMD type B are shown in Table 2 (pulmonary, hepato-splenic, and haematological assessments) and Table 3 (summary of all monitoring parameters). It is recommended that patient monitoring would be undertaken by a multidisciplinary team in an experienced centre (referral or competence centre in France). As with other diseases, the frequency of monitoring must be determined by patient severity and assessment results, as well as the progression of the disease.

Given the future availability of ERT, there is a need for biomarkers that predict or reflect disease progression [17]; suitable markers need to be validated for their use as surrogate markers of clinically relevant endpoints. Eskes et al. (2020) highlighted that the best evaluated potential biomarkers for ASMD are DLco, spleen volume, platelet count, LDL-C, liver fibrosis (measured with fibroscan) and lysosphingomyelin [17]. Our opinion is that the biomarkers chitotriosidase, lysosphingomyelin, and lysosphingomyelin 509 should be assessed at baseline, with follow up every 3 or 6 months for the first year, and then every year (Table 3).

Given that ASMD is an autosomal recessive inherited condition, carrier testing for at-risk relatives, prenatal testing for a pregnancy at increased risk, and preimplantation genetic testing are all possible, following the identification of a *SMPD1* pathogenic variant in an affected family member [43,44]. In general, there is no specific complication during pregnancy in patients with ASMD type B; although, episodes of bleeding have been reported. ASMD type A is the most severe form of ASMD with rapidly progressive neurovisceral manifestations and fatal issues with multisystem involvement. For ASMD type A, prenatal diagnosis is routinely accomplished by mutational analysis when an index case is identified in the family [43].

### 4.3. Patients with ASMD: Stratification by Disease Severity

In France, the exceptional use of proprietary medicinal products which do not have a marketing authorisation and are not the subject of a clinical trial can occur under the following three conditions: (1) specialties are intended to treat, prevent, or diagnose serious or rare diseases; (2) there is no appropriate treatment available on the market; and (3) their efficacy and safety of use are presumed in the state of scientific knowledge and the implementation of the treatment cannot be postponed [45]. This exceptional so-called ‘Temporary Use Authorisation’ (TUA) is delineated by specific criteria that have to be defined according to the results of the different trials and the criteria of the expected future marketing authorisation.

Where treatment is warranted based on disease severity, early access to olipudase alfa can be possible based on the following conditions: (1) the patient or person of trust must be informed of the exceptional and critical nature of this requirement; (2) specialties are intended to treat, prevent, or diagnose serious or rare diseases; (3) there is no appropriate treatment available on the market; (4) the patient cannot be included in a clinical trial; and (5) the effectiveness and safety of use of the prescribed treatment are not only presumed in the state of scientific knowledge and the implementation of the processing cannot be postponed.

In order to define the access criteria to the TUA procedure, the Therapeutic Evaluation Committee for Visceral Lipidoses (CETLv) in France convened a disease severity classification of patients with ASMD type B or A/B, i.e., via the Brassier–Lidove classification (Table 4). Worsening dyspnoea, low platelets <50 G/L and/or recurrent bleeding, biological and/or histological liver damage, and painful splenomegaly are key features of severe disease. Important clinical criteria in paediatric patients include a break in the growth curve or painful splenomegaly (Figure 3).

During nominative early access to olipudase alfa the Brassier–Lidove classification (CETLv classification) (Table 4) has been applied. Of 29 adult patients from the Croix Saint-Simon Hospital (Paris, France), two were deemed to be outside the early access application because of a limited life expectancy, two were in CETLv Group 1 and were too severely affected to participate in ASCEND trial (Table 5) (one patient with symptomatic thrombocytopenia <40 G/L and one patient with severe pulmonary failure awaiting lung transplantation), two were in CETLv Group 2 (both patients were participating in the ASCEND study), eight patients were in CETLv Group 3 (including two patients receiving psychotropic drugs), ten patients were in CETLv Group 4, and five patients in CETLv Group 5. Of this patient cohort, 10/29 (34.5%) patients were deemed to be eligible for olipudase alfa via ATU according to the CETLv classification. In the Referral Center for Inherited Diseases of Metabolism at the Hospices Civils of Lyon (Lyon, France), there have been two children treated since 2015 and 2018 as part of the pediatric clinical trial and two children treated as part of a nominative ATU (Group 1). Two adults have also been treated in the ATU for severe lung damage (Group 1).

## 5. Early Access Experience to ERT in France

Prior to full marketing authorisation, as of 14 October 2021, compassionate use of olipudase alfa was initiated in France in 4 nominative pediatric patients and 19 adult patients. A total of 14 physicians originating from 12 different cities (Lyon, Clermont-Ferrand, Bordeaux, Angers, Paris, Marseille, Quimper, Clichy, Strasbourg, Lille, Toulouse, and Caen) are currently treating ASMD patients, at the rate of about 1 to 5 patients per center. Of note, 16 out of these 23 patients were initiated with olipudase alfa by the co-authors of this work. Dose escalation (Figure 2) was conscientiously followed.

## 6. Discussion

In ASMD, as the lysosomal sphingomyelinase enzymatic defect is expressed ubiquitously throughout the organs, improvement in one tissue may be considered as representative of an overall improvement in non-neurological clinical status.

Available evidence from the natural history of ASMD and observational studies support the use of DLco and splenomegaly as clinically meaningful endpoints that assess disease burden for patients. Worsening DLco is a strong indicator of progressive lung disease in chronic visceral ASMD, contributing to increased disease burden. Likewise, spleen volume may be considered as a surrogate marker for bleeding risk and liver disease. Pneumonia/respiratory failure, and/or liver failure, are often the causes of death [9].

The main clinical criteria that were considered in France to prioritise early access to treatment with olipudase alfa in patients with ASMD type B or type A/B are low DLco with worsening dyspnoea, thrombocytopenia <50 G/L or recurrent bleeding, biological and/or histological liver damage, and painful splenomegaly (Figure 3), and in the case of paediatric patients a break in the growth curve or painful splenomegaly should be considered (Figure 3). Thus, a complete evaluation of the patient prior treatment initiation is mandatory in order to assess the potential benefit of olipudase alfa on the spleen volume, respiratory function, and platelet count, as well as any possible issues related to tolerance and comorbidities, such as tobacco use. In addition, as lung, spleen, liver, platelet status, and growth curve can worsen, a regular follow up with clinical, biological, and morphological of pauci/asymptomatic patients must be performed. The CETLv stratification of severity may aid the physician in health care decisions.

Data from the ASCEND (adult) and ASCEND-Peds (paediatric) studies have demonstrated that olipudase alfa can improve respiratory status and reduce spleen volume in patients with visceral ASMD (Table 5) [27,31]. In addition, the clinical relevance of the endpoints used in these studies is well documented [46]. Of note, patients included in the ASCEND (adult) and ASCEND-Peds studies did not exhibit the full spectrum of severity in patients with ASMD, excluding asymptomatic individuals and those with very severe disease. After dose escalation, steady-state dose infusions, and record of adequate tolerance to olipudase alfa, home therapy with a multidisciplinary team and regular follow-up should be considered. An ‘emergency card’ delivered to the patient could be helpful to improve the communication between the patient, and home and hospital caregivers in patients in home infusion.

The present early experience can be used as the basis for further clinical evaluations.

## 7. Conclusions

The management of chronic visceral and ASMD type B and A/B has traditionally been limited to supportive care and palliation. Clinical studies in patients with chronic ASMD have demonstrated positive results with the novel ERT olipudase alfa, comprising improvement in respiratory status and spleen volume reduction, as well as improvement in other clinical manifestations. In France, low DLco with worsening dyspnoea, thrombocytopenia <50 G/L or recurrent bleeding, biological and/or histological liver damage, and painful splenomegaly have been used to prioritise early compassionate use of olipudase alfa in nominative paediatric and adult patients with ASMD. Of note, the use of any decision-making process/stratification for patients with ASMD need to be confirmed by ‘real-life’ evidence. In addition, scheduled follow up of treated patients outside clinical trials is mandatory to assess the effect of olipudase alfa in those individuals with chronic ASMD. Regular follow up must be performed in untreated patients with ASMD. Asymptomatic patients should be evaluated by experts on a ‘case-by-case’ basis to ensure opportune early treatment initiation.

## Figures and Tables

**Figure 1 jcm-11-00920-f001:**
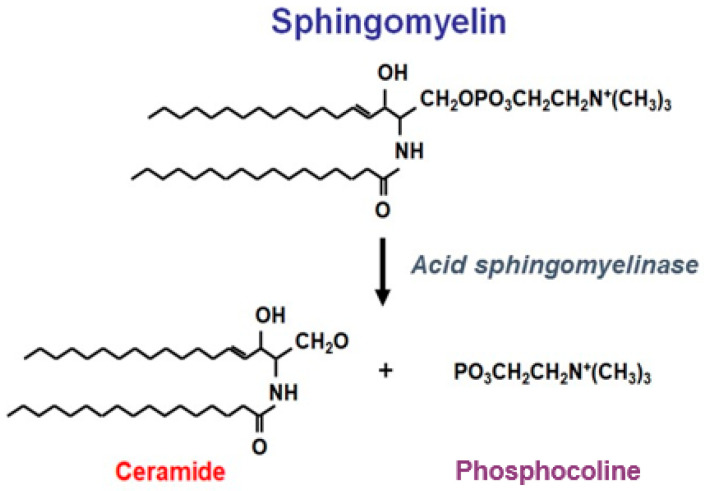
Enzymatic action of acid sphingomyelinase (ASM). Normal physiological function requires ASM to catalyse the hydrolysis of sphingomyelin to ceramide and phosphocoline. Ceramides have a proinflammatory effect as they are bioactive sphingolipids that act as second messengers which transduce pro-inflammatory signals.

**Figure 2 jcm-11-00920-f002:**
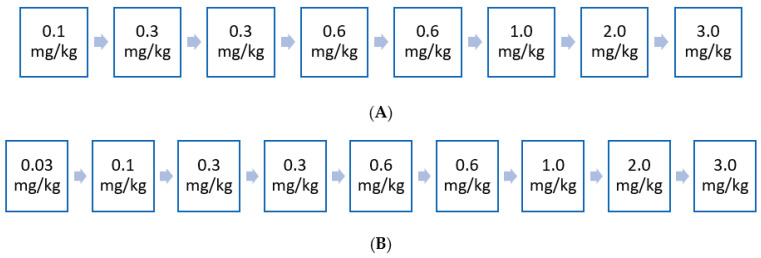
(**A**) Dose escalation regimen for adult patients administered every 2 weeks. Dose-escalation strategy for olipudase alfa after the Phase 1B study [31]. The aim of the dose-escalation process of olipudase alfa is gradual sphingomyelin debulking to prevent high ceramide release. BMI, body mass index. (**B**) Dose escalation regimen for pediatric patients administered every 2 weeks. Dose escalation regimen of olipudase alfa phase I/II ASCEND PEDS [27].

**Figure 3 jcm-11-00920-f003:**
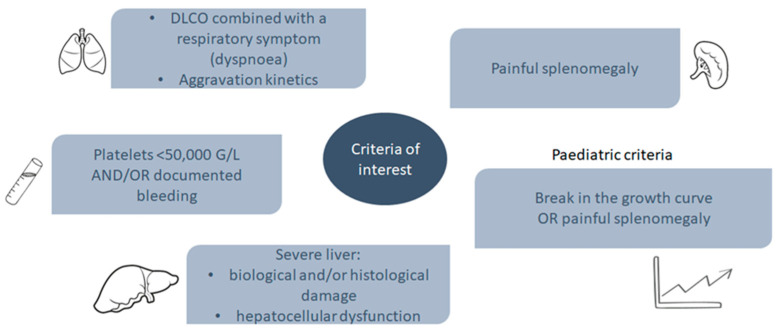
Clinical criteria of interest for patients with ASMD type B or A/B. ASMD, acid sphingomyelinase deficiency; DLco, diffusing capacity of the lungs for carbon monoxide.

**Table 1 jcm-11-00920-t001:** Nomenclature for ASMD, adapted from [7].

Historical Classification	Recommended Nomenclature
Niemann–Pick disease type A (NPD A)	Infantile neurovisceral ASMD (ASMD Type A)
Intermediate or variant phenotype (NPD A/B)	Chronic neurovisceral ASMD (ASMD Type A/B)
Niemann–Pick disease type B (NPD B)	Chronic visceral ASMD (ASMD Type B)

ASMD, acid sphingomyelinase deficiency; NPD, Niemann–Pick disease.

**Table 2 jcm-11-00920-t002:** Key monitoring parameters and frequency of assessments for patients with ASMD type B forms.

Exam	Baseline Visit	Follow-Up Visit in the First 6 Months	Follow-Up Visit Every 3 Months for the First Year	Follow-Up Visit Every Year
**Pulmonary assessment**				
Respiratory functional exploration (DLco—not before 6 years)	X	X		X
CT: thoracic	X			X ^a^
**Hepato-splenic** **assessment**				
Splenic and hepatic volume (ultrasound +/− abdominal MRI in adults; ultrasound in children)	X	X		X
Portal hypertension (abdominal ultrasound with Doppler)	X	X		X
Transaminases, alkaline phosphatase, bilirubin	X	X		X
**Haematological** **assessment**				
CBC, platelets, haemostasis assessment (PT, factor V, fibrin, APTT, ferritin)	X		X	X
Lipid blood-test including total, LDL and HDL cholesterol and triglycerides	X		X	X

^a^ If normal, then every 5 years. APTT, activated partial thromboplastin time; CBC, complete blood count; CT, computed tomography; DLco, diffusing capacity of the lungs for carbon monoxide; HDL, high density lipoprotein, LDL, low density lipoprotein; MRI, magnetic resonance imaging; PT, prothrombin time.

**Table 3 jcm-11-00920-t003:** Key monitoring parameters and frequency of assessment for patients with ASMD type B forms (NPD B).

Test	Monitoring Frequency	Additional Points
**Lung assessment**		
PFT		PFT possible in children from 5–6 years Need to monitor DLco (annually or at clinician’s discretion) following olipudase alfa treatment initiation
Blood gases	-	Non-predictive test Not routinely recommended, especially in children, and depending on symptomatology in adults
CT scan	At baseline visit and at follow-up visit annually, unless normal (then assess every 5 years)	Radiation examination in a population at risk of cancer In paediatric patients <4–5 years, sedation or general anaesthesia is required (depending on the child’s behaviour)
**Hepato-splenic** **assessment**		
Liver blood tests	At baseline visit, at follow-up visit in first 6 months, and at follow-up visit annually	Assess GGT, transaminases, ALP, bilirubin, PT, CRP, Factor V
Abdominal ultrasound with doppler	At baseline visit, at follow-up visit in first 6 months, and at follow-up visit annually	Early detection of steatosis, PHT, cirrhosis or nodule Possible in children < 5 years of age
Abdominal MRI (adult patients)	At initial assessment and then every 2 years	Higher sensitivity for nodule detection Possible in children > 5 years of age
Fibroscan	-	Not recommended as not validated for ASMD
Liver biopsy	To consider if hepatocarcinoma is suspected	No correlation with biological tests (transaminases usually < 5 N). Characterisation of nodule may require alpha-foetoprotein, liver ultrasound, and MRI
Alpha-fetoprotein	-	No recommendation for systematic assessment
**Haematological** **assessment**		
CBC, platelets, haemostasis test (PT, Factor V, fibrin, aPTT), ferritin	Assessment at baseline visit, at follow-up visit every 3 months during first year, and at follow-up visit annually	-
Protein electrophoresis	At initial assessment and then annually	Risk of hyper or hypogammaglobulinemia and risk of MGUS No need to perform any immunoelectrophoresis
Serum albumin	At initial assessment then every 2–3 years if the initial assay is normal	-
**Bone assessment and growth evaluation**		
Growth curve	Assessment at baseline visit and at follow-up visit every 6 months	-
Phosphocalcium balance	Assessment at baseline visit and at follow-up visit every 6 months	Assessment to include blood calcium and phosphorus, vitamin D, creatinine, proteinuria, urine calcium and sodium, and creatininuria Patients are at risk of cholestasis
Absorptiometry	Assessment at baseline visit and then every 5 years if normal or every 3 years if anormal	Possible in children aged from 5–6 years old
Resorption markers	Optional	To be performed if osteoporosis is known
**Cardiovascular assessment and lipid profile**		
Echocardiography	At initial assessment and then every 2 years	
Coronary computed tomographic angiography	Discuss coronary computed tomographic angiography depending on lipid profile and other cardiovascular risk factors	No monitoring data in ASMD type B disease
Lipid profile	Assessment at baseline visit, at follow-up visit every 3 months during first year, and at follow-up visit annually	Unproven benefit of statins in primary prevention
**Neurological assessment ^a^**		
*Peripheral*		
Clinical examination	Monitoring frequency: at initial assessment and then annually	
EMG	Monitoring frequency: to be considered if there are clinical call points	
*Central nervous system*		
Brain MRI	Monitoring frequency in adults: to be considered at initial assessment according to the clinical context and the neuropsychological tests Monitoring frequency in children: to be carried out at initial assessment if the diagnosis is made at a very early stage of childhood	
**Ophthalmic assessment**		
Ocular fundus	At initial assessment	The macular cherry-red spot is present in all patients with form A, and has been reported in one third of patients with form B [20]
Visual acuity	At initial assessment	Loss of visual acuity in infantile type (no effect in chronic visceral ASMD)
**Other**		
Dermatological examination	At initial assessment and then annually	Necessary in order to assess the presence of lymphoedema (eyelid infiltration)
TSH assay (in combination with anti-TPO antibodies)	At initial assessment and then annually	Necessary for the investigation of thyroid autoimmunity
**Biomarkers**		
Chitotriosidase	At initial assessment, follow-up every 3 months for the first year, and then every year	Chitotriosidase activity is absent in 6–8% of individuals in the general population
		Specialised assays, contact reference laboratories
Lysosphingomyelin + Lysosphingomyelin isoform 509	At initial assessment, follow-up every 3 months for the first year, and then every year	Specialised assays, contact reference laboratories

^a^ Note: The clinical dichotomy between forms A and B occurs very early before the age of 1 year (around 3–6 months). For form A/B, the age of onset of neurological signs is highly variable (delayed acquisition at a variable age, possible after the age of 3 years). aPTT, activated partial thromboplastin time; CBC, complete blood count; CRP, C-reactive protein; CT, computerised tomography; DLco, diffusing capacity of the lungs for carbon monoxide; EMG, electromyography; GGT, gamma-glutamyl transferase; MGUS, monoclonal gammopathy of unknown significance; MRI, magnetic resonance imaging; PFT, pulmonary function test; PHT, portal hypertension; PT, prothrombin time; TSH, thyroid-stimulating hormone; TPO, thyroid peroxidase.

**Table 4 jcm-11-00920-t004:** Patients with chronic ASMD grouped among disease severity (CETLv classification).

Group	Definition
Group outside the early access spectrum application	ASMD types B and A/B (NPD B or A/B) with a short-term vital prognosis (e.g., cancer)
Group 1: Patients with severe organ involvement	DLco ^a^ <50% and/ or dyspnoea Platelets <50 G/L ± recurrent bleeding or bruising Abdominal pain Urgent need to start treatment Break in growth curve (≥2 standard deviation)
Group 2: Patients in therapeutic trials	Patient continues to receive treatment as part of clinical trial or extension study
Group 3: Patients with moderate organ involvement	50% < DLco < 70% 50 G/L < Platelets < 100 G/L without bleeding or bruising Failure to thrive/break in growth curve (>1 standard deviation)
Group 4: Patients with mild organ involvement	DLco > 70% Platelets > 100 G/L Abnormal thoracic imagery and/or hepatosplenomegaly
Group 5: Patients with no symptoms	DL_CO_ > 70% Platelets > 100 G/L No symptom Normal thoracic imagery No hepato-splenomegaly

ASMD, acid sphingomyelinase deficiency; DLco, diffusing capacity of the lungs for carbon monoxide; NPD, Niemann–Pick disease. ^a^ DLco not available for patients aged <6 years old.

**Table 5 jcm-11-00920-t005:** Criteria in ASCEND (adults and peds) study for use of olipudase alfa.

Criteria	ASCEND (Adults)	ASCEND Peds
Lung	Inclusion criteria: DLco ≤ 70% Exclusion criteria: Invasive ventilation Non-invasive ventilation > 12 h/day	Exclusion criteria: Invasive ventilation Non-invasive ventilation > 12 h/day
Spleen	Inclusion criteria: Spleen volume ≥6 MN, measured by MRI; patients who have had a partial splenectomy may be included if the procedure was performed ≥1 year prior to screening and the residual spleen volume is ≥6 MN SRS score ≥ 5 over one week	Inclusion criteria: Spleen volume ≥5 multiples of normal (MN), measured by MRI; patients who have had a partial splenectomy may be included if the procedure was performed ≥1 year prior to screening and the residual spleen volume is ≥5 MN
Liver	Exclusion criteria: Active hepatitis B or hepatitis C ALT or AST > 250 IU/L or total bilirubin > 1.5 mg/dL (except for patients with Gilbert’s syndrome) Major organ transplantation (liver, bone marrow) The patient is unwilling or unable to refrain from taking any potentially hepatotoxic medication or herbal supplement 10 days before and 3 days after the liver biopsies	Exclusion criteria: Active hepatitis B or hepatitis C Cirrhosis (determined by clinical assessment) Major organ transplantation (liver, bone marrow) ALT or AST > 250 IU/L or total bilirubin > 1.5 mg/dL
Blood	Exclusion criteria: Platelet count < 60 G/L INR > 1.5 The patient is unwilling or unable to avoid taking any medication or herbal supplements that may cause or prolong bleeding, 10 days before and 3 days after the liver biopsies	Exclusion criteria: Platelet count < 60 G/L INR > 1.5
Bone/growth retardationBone damageDelayed growth	No mention of bone criteria Inclusion criteria: Size of −1 Z-score or lower	No mention of bone criteria
Central nervous system	No mention of criteria	Exclusion criteria: Acute or rapidly progressive neurological abnormality Homozygosity for the R496L, L302P, and fs330 mutations or any combination of these 3 mutations Delayed motor skills
Cardiovascular	Exclusion criteria: Significant heart disease Major organ transplantation (liver, bone marrow) No mention of dyslipidaemia/coronary calcifications	Exclusion criteria: Significant heart disease No mention of dyslipidaemia/coronary calcifications
Pain/fatigue	Inclusion criteria: SRS score ≥ 5	No mention of specific criteria

ALT, alanine aminotransferase; AST, aspartate transaminase; DLco, diffusing capacity of the lung; INR, international normalised ratio; MN, multiples of normal; MRI, magnetic resonance imaging. SRS (Splenomegaly Related Symptom) score: a score composed of 5 items (abdominal pain, abdominal discomfort, early satiety, self-image, ability to stoop). This score is derived from the Myeloproliferative Syndrome Assessment Score (MF-SAF) (JAKARTA, NCT# 01437787). The questions assess the effect over the last 24 h of the splenomegaly in patients with NPB disease.

## Data Availability

Not applicable.

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
