# Peer review of "Acid Sphingomyelinase Deficiency: Sharing Experience of Disease Monitoring and Severity in France"

_jcm, 2022, doi:10.3390/jcm11040920_

Round 1

Reviewer 1 Report

The manuscript by Mauhin et al. gives a nice review of the spectrum of disease, diagnosis and follow-up recommendations for ASMD. Some minor comments:

1. Line 127, p4: sphyngomyelin debulking --> sphingomyelin

2. Line  213, page 5: abdominal shock risks--> abdominal impact or trauma?

3. Table 3: the comments in the right column are confusing. Are you recommending echo/doppler for the detection of portal hypertension? Are you suggesting Fibroscan is not reproducible? It is unclear which comment comes with which exam. If the questions I ask here are answered with ‘yes’ then I’m afraid those comments need to be revised. In addition, splenectomy is not a test? Please redesign the Table for clarity and maybe even for evidence base?

Necessary in order to detect PHT
For children under 5 years of age
A more accurate test for nodule detection In children: possible from 5 years of age Not recommended as not reproducible Do not rely on normal functional tests
for confirmation (transaminases not very high [<5 times normal])
Not recommended Contraindicated as it may aggravate liver and lung damage

4. Line 276, page 9: Autorisation Temporaire d'Utilisation (ATU) – short explanation?

Major comment: the manuscript would benefit from a glimpse of the results in the "nominative treated patients". We suppose this will be the subject of a follow-up manuscript?

Reviewer 2 Report

Mauhin et al. described their own experience of diagnosis, treatment and monitoring of ASMD patients.

The paper is quite good, however I have some issues to be revised.

Verse 91: Please, define the term ,,hepatic dysfunction''. Does it refer to elevation of serum transaminases? What about acute liver failure (ALF) - are there reports of ALF in the course of ASMD in the literature? The accummulation in the reticuloendothelial system does not affect the hepatocytes function. 

Verse 200: Please, define the term ,,improving liver function''. Which parameters?

Verse 206: Are statins useful in the treatment of lipid abnormalities in ASMD? What is the pathogenesis of lipid abnormalities in ASMD? Do statins respond to these abnormalities? 

Verse 219: Please, cite the proper reference. What about thrombocytopenia as a possible contraindication to vaccination anti-SARS-CoV-2 and a possible risk for bleeding?

Table 2: (1) Why bilirubin should be monitored? Does ASMD cause cholestasis? Infantile chronic visceral forms of ASM deficiency could be a cause of cholestasis but what about adult patients? (2) Abdominal ultrasound with doppler imaging should be used to assess the possible development of portal hypertension.

Table 3: (1) ,,Liver biopsy - To consider if one or more nodules appear''. What nodules? Do You mean HCC development? Liver biopsy should be avoided in ASMD patient due to lack of indications. AFP, liver ultrasound (and MRI) are useful for possible HCC development. (2) Cherry red spot was also described in Chronic visceral ASMD (not only neuronopathic forms).

Paragraph 4.3 - What about patients with SMPD1 polymorphism?

Reviewer 3 Report

  • In dose escalation studies Phase 1B, what is the highest ceramide leves in adult and paediatric patients. When compared with Phase 1 trial what is the statistical significance? 
  • Why A/B patients are selected for ERT since this will not influence neurological symptoms.
  • What is the proportion of cases with A, B, and A/B? Among the B cases, what proportion are symptomatic?
  • What is your recommendation with regard to the dosage? Do you recommend Phase escalation of the dose?

Round 2

Reviewer 2 Report

The Authors responded to all my questions.

The manuscript was revised according to my suggestions.